# IL-5 Serum and Appendicular Lavage Fluid Concentrations Correlate with Eosinophilic Infiltration in the Appendicular Wall Supporting a Role for a Hypersensitivity Type I Reaction in Acute Appendicitis

**DOI:** 10.3390/ijms232315086

**Published:** 2022-12-01

**Authors:** Nuno Carvalho, Elisabete Carolino, Hélder Coelho, Ana Cóias, Madalena Trindade, João Vaz, Brigitta Cismasiu, Catarina Moita, Luis Moita, Paulo Matos Costa

**Affiliations:** 1Serviço Cirurgia Geral, Hospital Garcia de Orta, 2805-267 Almada, Portugal; 2Faculdade Medicina, Universidade Lisboa, 1649-028 Lisboa, Portugal; 3H&TRC-Health & Technology Research Center, ESTeSL-Escola Superior de Tecnologia da Saúde, Instituto Politécnico de Lisboa, 1990-096 Lisboa, Portugal; 4Serviço de Anatomia Patológica, Hospital Garcia de Orta, 2805-267 Almada, Portugal; 5Innate Immunity and Inflammation Lab, Instituto Gulbenkian de Ciência, 2780-156 Oeiras, Portugal; 6Instituto de Histologia e Biologia do Desenvolvimento, Faculdade Medicina, Universidade Lisboa, 1649-028 Lisboa, Portugal

**Keywords:** allergy, appendicitis, appendicular lavage fluid, eosinophils, IL-5, Th2 cytokine

## Abstract

Appendicitis is the most common abdominal surgical emergency, but its aetiology is not fully understood. We and others have proposed that allergic responses play significant roles in its pathophysiology. Eosinophils and Interleukin (IL)-5 are involved in a hypersensitivity type I reaction. Eosinophil infiltration is common in the allergic target organ and is dependent on IL-5. In the presence of an allergic component, it is expected that the eosinophil count and IL-5 local and systemic concentrations become elevated. To address this hypothesis, we designed a prospective study that included 65 patients with acute appendicitis (grouped as acute phlegmonous or gangrenous according to the histological definition) and 18 patients with the clinical diagnosis of acute appendicitis, but with normal histological findings (control group) were enrolled. Eosinophil blood counts and appendicular wall eosinophil infiltration were determined. IL-5 levels in blood and appendicular lavage fluid were evaluated. Appendicular lavage fluid was collected by a new methodology developed and standardized by our group. Appendicular wall eosinophil infiltration was higher in acute phlegmonous appendicitis than in gangrenous appendicitis (*p* = 0.000). IL-5 blood levels were similar in both pathologic and control groups (*p* > 0.05). In the appendicular lavage fluid, the higher levels of IL-5 were observed in the phlegmonous appendicitis group (*p* = 0.056). We found a positive correlation between the appendicular wall eosinophilic infiltration and the IL-5 concentrations, in both the blood and the appendicular lavage fluid, supporting the IL-5 reliance in eosinophil local infiltration. We observed the highest presence of eosinophils at phlegmonous appendicitis walls. In conclusion, the present data are compatible with a hypersensitivity type I allergic reaction in the target organ, the appendix, during the phlegmonous phase of appendicitis.

## 1. Introduction

Acute appendicitis (AA) is the most frequent abdominal emergency in general surgical practice worldwide [1]. The role of several putative aetiologic factors is still misleading: namely, the interaction between >predisposing and precipitating factors remains poorly understood [2]. Some contemporary insights suggest that allergy may be an important player in AA [3,4]. Recent and ongoing results of our group corroborate this line of research [5,6] and allergy was searched for as the “missing link” in the aetiology of AA [7]. If, in this narrative revision [7], a definitive causal relation between allergy and acute AA could not be established, it was well stated that allergic features are undoubtedly present in AA [5,6,7,8,9]. Addressing the allergic component of AA is a field of interest for future research that may sustain recent changes in clinical decision-making [5,10,11].

Histological findings suggest that AA may be an allergic reaction [3,4]. Features of hypersensitivity type I reaction, such as the presence of eosinophil infiltration, mast cell degranulation, oedema of germinative centres, perivascular histiocytic proliferation, and muscular oedema, are common in appendicular specimens of AA [3,4].

Clinical and experimental data clearly illustrate allergic reactions, or an immune component of it, in the context of AA. Atopic patients are more prone to AA than non-atopic individuals [8]. Patients with asthma have increased risk of appendectomy [12]. A lower risk of complicated appendicitis was reported in children with IgE-mediated allergy, suggesting that immunologic disposition modifies the clinical pattern of appendiceal disease [9]. Increased allergic reactions are reported in patients with appendicitis, as assessed by a skin prick test [8].

Allergy, expressed as an immune response to environmental antigens (allergens) that involve Th2 cells, mast cells, eosinophils, and immunoglobulin E (IgE) [13], is a suitable trigger for AA, as the digestive tract is challenged by thousands of antigens daily and the immune system can react against otherwise harmless environmental agents [8,14].

Type 2 immune responses are expressed by the cytokines, interleukin (IL)-4, IL-5, IL-9, and IL-13, acting either as host-protective or as pathogenic inducers [15]. When type 2 responses are dysregulated, they can become important drivers of disease [16], such as in eosinophilic esophagitis [8].

IL-5 is the primary cytokine involved in the recruitment, activation, and survival of eosinophils [17]. An elevation of IL-5 and eosinophil infiltration at target organs is common in allergic reactions [18,19]. Eosinophils are end-stage effector cells involved in the pathogenesis of Th2 immune-mediated disorders and eosinophilia is a hallmark of allergies [20].

Recently, we published the development of the concept of appendicular lavage fluid (ALF) to evaluate local immune response in AA [5]. An elevation of Th2 cytokine profile (IL-4, IL-5, and IL-9) was documented in the ALF of phlegmonous appendicitis [5]. Furthermore, using a monoclonal antibody anti-IgE, we found high levels of IgE fixation in appendicular specimens of phlegmonous appendicitis [6]. These data corroborate the interest for exploring the allergic component in phlegmonous appendicitis, foreseeing a possible incorporation of the incoming results in the clinical reasoning. Will appendicitis be another digestive manifestation of a Th2 type allergy response? [8]

The aim of this study was to evaluate organ manifestation of an allergic reaction in the wall of the appendix (eosinophil infiltration) through sequential phases of AA and search for a possible correlation with IL-5 present in ALF and in the blood.

## 2. Results

The study population was homogeneous in terms of age, gender, history of allergies, and body mass index (Table 1).

Eosinophil blood counts ranged from a minimum of zero in 10 patients (5 APA, 4 AGA, and 1 NPA) to a maximum of 0.5 × 10^6^ mL in the NPA, with a median blood level of 0.06 and interquartile range of 0.02–0.14. A significant statistical difference was found in the eosinophil blood counts in the three histologic groups (*p* = 0.004) (Table 2) (Figure 1A).

Pair-wise analysis showed differences between the pair NPA–AGA (*p* = 0.007), where the highest number of blood eosinophils occurred in NPA. The difference between the pair APA–AGA was significant (*p* = 0.04), the highest number being associated with APA.

At the appendicular wall, the infiltration of eosinophils was not found in the same proportions as that found in the blood (Table 2).

For the appendicular wall, the higher values were associated with APA, followed by NPA and AGA (*p* = 0.000) (Figure 1B).

The median eosinophils number in the appendicular wall was 41.00, with an interquartile range of 21.00–70.00.

IL-5 was detected in the blood of all the patients, ranging from a minimum of 5.029 pg/mL in a patient with APA to a maximum of 203.9 pg/mL in a patient with AGA (median = 1.81 pg/mL and interquartile range = 10.57–19.67 pg/mL). No difference was found between IL-5 blood levels among the histologic groups (*p* = 0.239) (Table 2) (Figure 1C).

IL-5 levels in the ALF were detected in all histologic groups (median = 16.67 pg/mL and interquartile range = 7.33–46.67 pg/mL)—Table 2 (Figure 1D). The mean distribution of IL-5 in ALF mimics the IL-5 distribution in blood, although no correlation was found between IL-5 blood levels and IL-5 ALF levels (*p* = 0.440) when they were compared by histology groups (Table 3). A marginal statistical difference was found for IL-5 ALF levels, with the highest levels in APA (*p* = 0.056).

Significant correlations were detected between peripheral blood eosinophil counts and appendicular wall eosinophil counts (*p* = 0.009) (Table 3) (Figure 1D).

No correlation was found between IL-5 blood levels and IL-5 ALF levels (*p* = 0.440) (Table 3).

A significant correlation was also detected between IL-5 levels, in blood (*p* = 0.046), lavage fluid (*p* = 0.018), and the eosinophil appendicular wall count (Table 3). For IL-5 levels, the most striking correlation was found between ALF levels and eosinophil appendicular wall infiltration in APA (*p* = 0.019) (Figure 2).

## 3. Discussion

Lower abdominal pain in the right quadrants was a widespread clinical situation and the clinical reasoning for the differential diagnosis has been well structured and established accordingly. In the “early” years, the main aetiologies were suspected by clinical details and to operate-or-not-to-operate was challenging (appendicitis vs. acute non-specific abdominal pain) [2,21]. In females, other diagnoses came up often. The supposed sequence of events was normal appendix, phlegmonous appendicitis, gangrenous appendicitis, perforation, and peritonitis was “la bête noire”—remove the appendices as soon as AA was diagnosed, and prevent abscess formation and sepsis, was the state of the art for many generations of surgeons [2,22,23]. Scores for predicting the state of the evolution of patients’ conditions were founded on clinical, lab, and image data [2,21,22,23,24].

In the beginning of this century, a strong trend to not operate on all the patients with appendicitis gained traction [10,25,26,27]. The analysis of the results of this policy is out of the scope of this paper. During the pandemic of COVID-19, a widespread policy of operative restriction enlarged our knowledge on the attempts not to operate unless the risk of infection spread was high [28,29]. Regardless of the proposed change in guidelines, this novel approach opens the field of research on appendicitis models of aetiology and treatment [10,11]. As an example of these emerging concepts, the role of allergy as a trigger or a cofactor in some patterns of AA is being explored.

As the appendix is a lymphoid organ, the possibility of an immune response to an unrecognized local antigen was considered [29]. Allergy, an allergen-specific hypersensitivity reaction type I [9], may flourish in the appendicular ground, as the microbiome and thousands of antigens are daily challenging the immune system to react against otherwise harmless environmental agents [8,14].

Although patients with AA are more commonly atopic than the general population [8,12], in our study group, no difference was found regarding the presence of allergy between patients with AA and the control group. Allergy is a complex event. Frequently, people with allergies have more than one allergic disease, but the presence of just one allergic disease is not unusual. Furthermore, the first manifestation of an allergy can be, for example, an anaphylactic reaction, and so, acute appendicitis can be the first allergic manifestation with clinical relevance to an unknown antigen.

Eosinophil infiltration, mast cell degranulation, and muscle oedema are features of hypersensitivity type I reaction that are common in AA [4]. The Th2 cells, through IL production, are responsible for a type I hypersensitivity reaction, with eosinophils being involved via IL-5, which controls the recruitment and function of eosinophils in target tissues [18,30,31]. Eosinophils are implicated in the initiation and maintenance of type 2 immune responses [32]. They are formed in the bone marrow, released into circulation, and accumulate in the tissue, where they synthesize mediators that cause tissue damage [20,32].

In most allergic diseases, such as nasal rhinitis, nasal polyps, asthma, atopic dermatitis, and eosinophilic esophagitis, a pronounced eosinophilic infiltration of organs is found [18,33].

In this study, the eosinophil count in the appendicular wall of the appendicular specimen’s histology was significantly higher in APA. The lower number of eosinophils was detected in AGA. When we analysed the presence of IgE in the walls of AA specimens, a similar distribution of intensity was reported [6].

The putative contribution of IgE to an allergic reaction was more intense in phlegmonous appendicitis. A lower number of IgE staining cells in gangrenous appendicitis could be due to tissue destruction [24].

The intensity of eosinophilic infiltration of the appendicular wall was correlated with systemic IL-5 levels and with IL-5 levels in the liquid collected by ALF procedure. Actually, this positive correlation found between the IL-5 levels in blood and ALF with the eosinophil infiltration of the appendicular wall may reflect the eosinophils’ dependence on IL-5 for migration to tissues such as the appendix [34].

In healthy subjects, IL-5 serum levels may be under the lower limit of detection, either because the levels are very low or because they are not produced at all [35]. In our study, IL-5 was detected in all the blood samples, as well as in ALF, which can be interpreted as the expression of the Th2 cytokine allergic reaction [35]. Even in NPA cases, IL-5 was detected, namely, in ALF.

In previous works of our group [5], we showed that Th2 cytokine levels in ALF were higher in acute appendicitis. When comparing phlegmonous appendicitis values with those of NPA, the difference was significant (*p* = 0.01): IL-4 (48.3 vs. 21.3 pg/mL), IL-5 (29.2 vs. 8.0 pg/mL), and IL-9 (34.1 vs. 16.6 pg/mL). Our data support the hypothesis initially proposed by others [36], based on a small number of cases, that “patients developing gangrenous appendicitis could, hypothetically have a stronger and uncontrolled Th1 response compared to those with phlegmonous appendicitis” while the Th2 response profile was involved in modulating phlegmonous appendicitis [5,36]. The ALF could not completely reflect the situation in vivo but, in analogy with bronchoalveolar lavage fluid, we assume that it reflects the inflammatory environment of the appendix [5].

In cases of NPA and clinically suggestive of AA, the presence of a histologically normal appendix does not mean that an inflammatory or allergic reaction is absent. In fact, a normal appendix can show focal inflammation if serial sections are performed [29] and evidence of an inflammatory molecular response in the form of increased cytokine expression can be found in macroscopic normal appendix [37]. This means that NPA can, in fact, “hide” an acute appendicitis or that the appendix was removed in an early phase of the disease. So, it is possible that patients with pain in the right iliac fossa and no histologic criteria of AA are at an initial phase of AA and, as neutrophil infiltration has not yet occurred, the histological diagnosis of AA cannot be made. This can explain the presence of detectable IL-5 levels in patients with NPA that we are reporting. It is possible that if the inflammatory process will progress, the neutrophil muscular infiltration will be present over time and the diagnosis of AA will be made from histologic criteria. In a series of 214 acute appendicitis, studied by us, the lowest levels of eosinophils were found in the AGA and the highest in those with a normal appendix [38]. We reproduced these results here.

It is well-known that eosinopenia is common is sepsis [39]. In acute diverticulitis, there is a reduction of eosinophils from the Hinchey stage I to stage IV [40]. Bacterial overgrowth in the appendix occurs at late stages in AA progression [41]. The presence of infection can induce eosinophil blood count reduction [38,39,40,41]. Eosinopenia involves suppression of production and egress of mature eosinophils from the bone marrow, together with sequestration of circulating eosinophils [39,40]. Sequestration could be ascribed to the migration of eosinophils into the inflammatory site itself, which is mediated by chemotactic substances [34]. It would be interesting to know if there is an increase of eosinophils in the terminal ileum or cecum, reflecting more extensive inflammatory changes and not just a local process, but to perform colonoscopy with biopsies would raise ethical issues.

Is the eosinophil infiltration in the appendix wall the result or the cause of appendicitis? In allergies, the local release of IL-5 acts as a chemotactic factor for eosinophils to invade the target organ, as could be the appendices. Cytokines act as a trigger for appendicitis. Eosinophils release several cationic proteins, such as eosinophil cationic protein, eosinophil peroxidase, or eosinophil derived neurotoxin, that have powerful inflammatory and cellular destructive properties [42]. It seems that eosinophils at the appendicular wall may act as a promotor of the pathologic consequences that are characteristic of the different phases of appendicitis.

In the present study, eosinophil infiltration at the inflammatory site, the appendix, was higher in APA, and correlated with IL-5 levels, the chemotactic factor, both in blood and ALF. We can speculate that the progression of the inflammation of the appendix, culminating with bacterial invasion, leads to a reduction of the eosinophil blood count. At the appendicular wall of the target organ, the appendix, the eosinophils’ highest number are associated with APA. IL-5 ALF’s highest levels were present in APA. These findings corroborate the presence of a possible allergic component in APA. A hypersensitivity type I reaction can therefore be added as another factor involved in AA aetiology [7,24].

Our results point to a meaningful role of the type I allergic reaction in appendicitis, and it is probably involved in the early stage of appendicitis in a substantial number of cases. Most probably, the aetiology of acute appendicitis is multifactorially triggered by different conjugations of the appendix ecosystem. Currently, we are investigating other factors involved in type I hypersensitivity reactions and AA.

Strengths: The prospective nature of the study, with sound histologic confirmation of APA, AGA, NPA, and with precise criteria. To the best of our knowledge, this is the first study that specifically evaluates IL-5 in the context of AA using an innovative and unique methodology, the ALF, in order to study local inflammation and the relation with eosinophil infiltration in the appendicular wall specimens.

Limitations: The small sample size from a single institution. Lack of eosinophils and IL-5 blood concentrations before clinical appendicitis, as this would serve as a reference and will enable us to follow the dynamic of blood changes. The incompleteness of the data concerning other allergy manifestations means that the results should be revaluated and expanded by others.

## 4. Material and Methods

### 4.1. Study Population

This prospective single-centre design study enrolled patients admitted at the Hospital Garcia de Orta emergency department, between April 2016 and June 2017. We carried out a pilot study, as the current literature does not provide data for the calculation of the sample to be used [43]. The study population was composed of patients submitted to appendicectomy for the clinical diagnosis of AA. According to the histological report, three groups were generated: Acute Phlegmonous (Suppurative) Appendicitis (APA), Acute Gangrenous Appendicitis (AGA), and the control group if negative for appendicitis (Non-Pathological Appendix—NPA). The number of patients in each group was: APA = 37, AGA = 28, and NPA = 18.

All patients received intravenous general anaesthesia and perioperative antibiotic treatment.

Patients under 18 years of age were not included because these patients are taken care by the paediatrics department. Pregnant women were excluded as pregnancy can induce alterations in the humoral immune response [31].

Setting: the study was carried out in a tertiary public hospital, with a capacity of 600 beds, in an urban area, providing medical care to 280,000 patients.

The Hospital is affiliated with the Faculdade Medicina da Universidade de Lisboa for teaching and research.

### 4.2. Ethical Considerations

The study was approved by the Ethical Commission of Hospital Garcia de Orta (June 2015, Reference 05/2015 and June 2015, Reference 05/2015 Addendum I) and conducted in accordance with the Helsinki Declaration (World Medical). Written informed consent was obtained from the patients. All personal identification data were anonymized and de-identified before analysis.

### 4.3. Appendicular Lavage Fluid (ALF)

The appendicular specimen was removed and prepared for ALF samples. At the proximal lumen of the resected appendices, a gauge was inserted, according to the calibre of the appendicular lumen, and 3 cc of 0.9% saline was introduced and collected. The process was repeated 3 times and the strict protocol was performed exclusively by one of the authors (N.C.), to ensure that the conditions were standardized. A Sarstedt Monovette tube was used for the ALF samples collected. One mL of ALF supernatant was then extracted and stored at −20 °C.

These samples were processed as expressed in point 5.

### 4.4. Pathologic Analysis

The appendices surgically removed in the clinically suspected cases of AA were processed for ALF analysis and then placed in 10% formalin for histopathological examination. Appendicular tissue was fixated for at least 24 h. The appendicular specimen was cut in three pieces, respectively, tip, intermediate length, and base. One section was taken for each of the pieces and was fixed and paraffin embedded. Each block was cut in two sections of 5-micron thickness that were stained with haematoxylin and eosin [44].

The slides were visualized by light microscopy (“Nikon^®^ ECLIPSE E400”, Novel Optics Jiangnan, Nanjing, China). Eosinophils were identified as large cells with a bilobated nucleus and large acidophilic specific granules that either had a bright red or reddish-purple stain (Figure 3).

In order to count eosinophils, we evaluated 10 high power fields (objective 40×—22 mm diameter), 5 in a clockwise direction around the mucosa and submucosa, to guarantee that we would not count twice the same field, and 5 in a radial direction from the *muscular propria* to the serosa [6].

All surgical specimens were observed and classified by one of the authors (C.H.) as APA, AGA, and negative for appendicitis, NPA. The samples with neutrophil infiltrate in the *muscular propria* were classified as APA; necrosis of the appendix, with loss of cellular structure and nuclei, was classified as AGA. When no neutrophil infiltration was present at the appendicular wall *muscularis propria*, the specimen was considered as non-pathologic appendices, classically classified as negative for appendicitis [45,46,47].

Neutrophils at the mucosa and submucosa were considered as a normal variant with no clinical relevance [48,49].

No cases of appendicular specimens with parasitic infestation were noted in this study.

### 4.5. IL-5 Determinations

Blood samples of 5 mL were collected by venepuncture with a Sarstedt Monovette tube and centrifuged. An amount of 1 mL of the supernatant was extracted and stored at −20 °C.

IL-5 was evaluated in samples collected by ALF procedure.

The ELISA protocol was used for IL-5 determinations (Human IL-5 MAX standard set protocol, Biolegend, San Diego, CA, USA), according to the manufacturer’s protocol.

The IL-5 was expressed in pg/mL. The lower threshold for detection of IL-5 with this method was 4 pg/mL.

### 4.6. Eosinophil Blood Count Determination

White blood cell count was made on a DxH 900 equipment (Beckman Coulter, Inc.Brea, CA, USA), using the Coulter principle. For evaluating differential white blood cell populations, VCSn technology was used, and eosinophils were automatically expressed in mm^3^ in peripheral blood. The count was made in triplicate for assuring security and reproducibility of the results.

Reference values for eosinophils were 0.00–0.80 × 10^6^/mL (0.0–7.0%).

### 4.7. Other Data

Personal information including age, sex, comorbidities, symptoms onset and their duration, absence of peritonitis, localized or generalized peritonitis, surgical details, open or laparoscopic appendectomy, complications, hospital length of stay, and other histologic features were also evaluated (some data was not shown). Individuals were also enquired about any symptoms that could be related to allergic disorders.

### 4.8. Statistical Analysis

Statistical software SPSS, V27.0 for Windows was used for data analysis. A 5% significance level was considered as significant. The Shapiro–Wilk test was used for testing the normality of the present data. For qualitative data, frequency analysis (n, %) was performed for sample characterization; for quantitative data, median (percentile 25%–percentile 75% (Q1–Q3)) were used. To compare age and BMI, eosinophil blood count, eosinophil count in the appendicular wall, and the IL-5 blood levels between the appendicular specimen’s histology (the three study groups NPA, APA, and AGA), the Kruskal–Wallis test was used, since the normality assumption was not verified and given the discrepancy in the size of the groups. When statistically significant differences were detected, Kruskal–Wallis multiple comparison tests were used. To verify whether the distribution of gender and allergies (presence/absence) was homogeneous among the three groups under study, the Chi-Square test and the Chi-Square test by Monte Carlo simulation were used (since the assumptions of applicability of the Chi-Square test were not verified), respectively. The Spearman correlation test was used to measure the degree of association between two variables.

## 5. Conclusions

Th2 cytokine IL-5 is involved in allergic reactions, recruiting eosinophils to the target organ. In AA, IL-5 levels are elevated simultaneously in appendicular lavage fluid and serum and associated with eosinophil infiltration of the appendicular wall, suggesting the presence of a hypersensitivity type 1 reaction in AA. Eosinophils release several cationic proteins that induce local inflammation and destruction—appendicitis.

The highest concentrations of Th2 cytokine IL-5, in appendicular lavage fluid, were detected in APA. The concentration of this hypersensitivity biomarker was correlated with eosinophil infiltration in the appendicular wall. These data support the hypothesis of a local allergic component, hypersensitivity type I reaction, in acute phlegmonous appendicitis. Allergy can be a role-player in the pathogenesis of appendicitis, mainly during the phlegmonous phase.

## Figures and Tables

**Figure 1 ijms-23-15086-f001:**
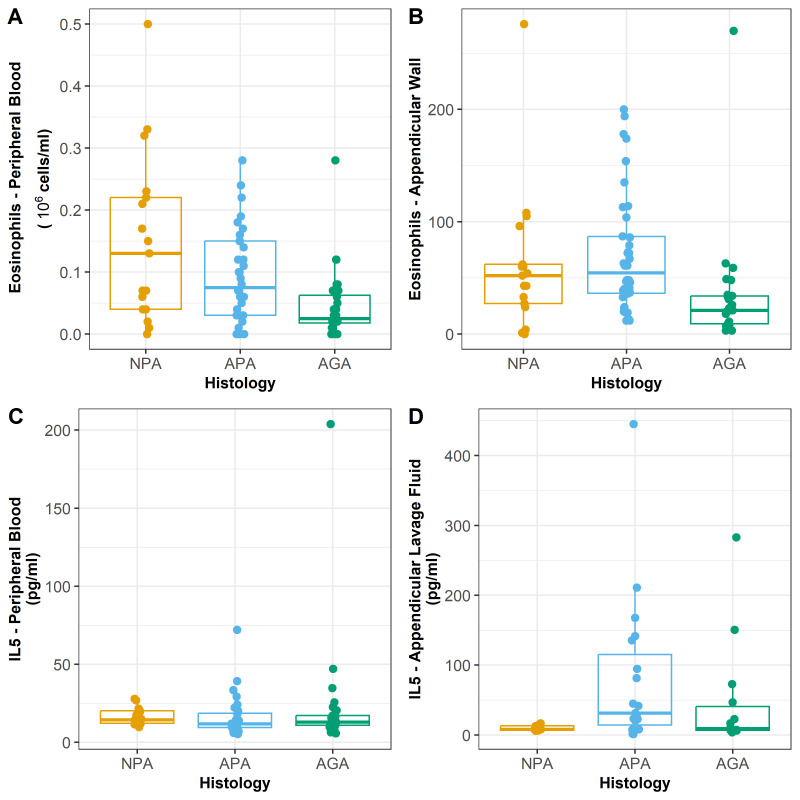
Boxplot summarizing IL-5 appendicular lavage fluid (ALF) in histologically proven normal appendix (NPA), acute phlegmonous (APA), and gangrenous appendicitis (AGA) groups. Median values, interquartile ranges, and ranges (excluding outliers and extreme values) are denoted by horizontal bars and boxes. The statistical analysis was carried out with independent samples Kruskal–Wallis test. (**A**) Blood eosinophils and histology. Eosinophils are presented in absolute number ×10^6^. (**B**) Eosinophils in appendicular wall and histology. Eosinophils are presented in absolute number. (**C**) Peripheral IL-5 and histology. IL-5 is presented in pg/mL. (**D**) Appendicular lavage fluid IL-5 and histology. IL-5 is presented in pg/mL.

**Figure 2 ijms-23-15086-f002:**
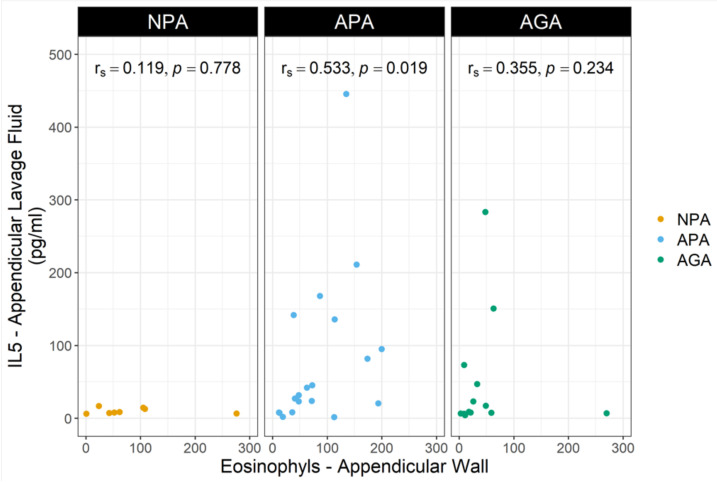
Scatter plot for the study of the relationship between IL-5 in the appendicular lavage fluid and the eosinophils in the appendicular wall in the control group (NPA), acute phlegmonous appendicitis (APA), and acute gangrenous appendicitis (AGA). The statistical analysis was carried out with Spearman correlation test.

**Figure 3 ijms-23-15086-f003:**
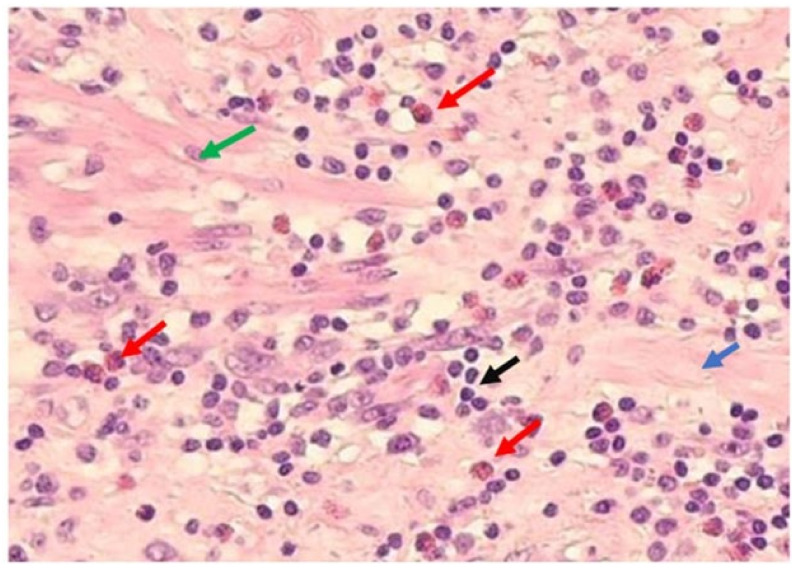
Appendicular section from appendix stained with haematoxylin and eosin: 200×, showing transmural inflammation. Eosinophils are presented as large cells with bilobated nucleus and large acidophilic specific granules (red arrow). Surrounding connective tissue (blue arrow), composed with myofibroblasts (green arrow), and inflammatory cells (black arrow).

**Table 1 ijms-23-15086-t001:** Patient’ Demographics and appendicular specimens’ pathology.

	NPA	APA	AGA	
N (%)	18 (21.7)	38 (45.8)	27(32.5)	*p* Value
Age (y)	31.5(27–40)	37(24–48)	38(26–48)	0.611 *
Gender (M/F)	14/4	25/13	14/13	0.196 **
Allergy (N/Y)	15/3	30/6	23/4	0.999 ***
BMI	22.7(21.40–25.85)	24.5(22–27.80)	25.30(22.30–28.60)	0.237 *

NPA—Non-Pathological Appendice; APA—Acute Phlegmonous Appendicitis; AGA—Acute Gangrenous Appendicitis; M—Male; F—Female; N—No; Y—Yes; BMI—Body Mass Index. Kg/m^2^. Results are presented as Median (Q1–Q3); * Kruskal-Wallis test. ** Qui-Square test. *** Qui-Square test by Monte Carlo Simulation.

**Table 2 ijms-23-15086-t002:** Eosinophils and IL-5 Levels.

	NPA	APA	AGA	
				*p* Value
Eosinophils (PB)	0.13(0.004–0.22)	0.08(0.03–0.15)	0.03(0.02–0.07)	0.004 *
Eosinophils (AW)	47.5(24–62)	1(37.5–95.5)	21(9–34)	0.000 *
IL-5 (PB)	14.38(12–20.52)	11.67(9.43–18.00)	13.05(10.76–18.79)	0.239
IL-5 (ALF)	8(6.67–13.6)	29.22(9.06–115.16)	8.67(6.67–46.67)	0.056

NPA—Non-Pathological Appendice; APA—Acute Phlegmonous Appendicitis; AGA—Acute Gangrenous Appendicitis. PB—Peripheral Blood Levels; AW—Appendicular Wall; ALF—Appendicular Lavage Fluid; Eosinophils (BL) presented in absolute number × 10^6^/ mL. IL-5 presented as pg/mL; Eosinophils (AW) presented in absolute number—Ten high power fields (objective 40×—22 mm diameter); Results are presented as Median (Q1–Q3); Kruskal-Wallis test. * Statistical Significant differences at a 5% significance level.

**Table 3 ijms-23-15086-t003:** Relationship of blood and wall eosinophils with IL–5 from PB and ALF.

	Eosinophiles—Wall	IL5—PB (pg/mL)	IL5—ALF (pg/mL)
Eosinophiles—PB	0.291 ^**(a)^	0.161	0.238
Eosinophiles—Wall		0.246 ^*(b)^	0.374 ^*(c)^
IL5—PB (pg/mL)			0.139 ^(d)^

PB—Peripheral Blood; ALF—Appendicular Lavage Fluid; * Correlation is significant at the 0.05 level (2-tailed); ** Correlation is significant at the 0.01 level (2-tailed); *p*-value (a) 0.009 (b) 0.046 (c) 0.018 (d) 0.440; Spearman’s correlation coefficient.

## Data Availability

The data of this study are restricted by the ethics committee of Hospital Garcia de Orta, to protect patient privacy. Data are available from Dr. Nuno Carvalho (nunomdcarvalho1964@gmail.com), for researchers who meet the criteria for access to confidential data.

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
