# Peer review of "IL-5 Serum and Appendicular Lavage Fluid Concentrations Correlate with Eosinophilic Infiltration in the Appendicular Wall Supporting a Role for a Hypersensitivity Type I Reaction in Acute Appendicitis"

_ijms, 2022, doi:10.3390/ijms232315086_

Round 1
Reviewer 1 Report
it is a well descriBED and interesting manuscript. the only problem i want to mention is the small amount of patients. i think other studies could prove this hypothesis.
Author Response
Review 1
it is a well described and interesting manuscript. the only problem i want to mention is the small number of patients. i think other studies could prove this hypothesis.
Thank you for the commentary
Sample size is often calculated to collect data that when statistically analysed will produce conclusions strong enough to be generalized. In statistical terms, is calculated in order to have a good statistical power to conclusively test the original hypothesis. In these calculations, we need the significance level (normally defined at 0.05), the desired power level and the effect size, ie, expected differences among group. Because, our study is the first of its kind, there was no prior data or expectations on these differences (either if they exist or their magnitude). It was then difficult, or even impossible, to determine a good sample size.
Currently we are working on other potential allergic involvement in acute appendicitis.
Reviewer 2 Report
This manuscript investigated the relationship between allergic component of serum and appendicular lavage fluid IL-5 levels, eosinophilic infiltration at appendicular wall and the aetiology of appendicitis and concluded that the hypothesis of a local allergic component, hypersensitivity type I reaction, in acute phlegmonous appendicitis. It is very interesting, but there were some major comments.
1. If the occurrence of acute appendicitis is related to type I allergy, why only focus on the study of IL-5 level?
2. If acute appendicitis is caused by allergy, why does the patient have no other allergic symptoms?
3. Is eosinophil infiltration in the appendix wall the result or cause of appendicitis?
4. Number of eosinophils in appendicitis was increased. Is there any increase of eosinophils in the terminal ileum or cecum?
5. The lavage solution of isolated appendix cavity by surgery could not completely reflect the situation in vivo.
6. According to the experimental design of this article, it is difficult to conclude that the occurrence of type I allergic reaction was the aetiology of acute appendicitis.
Author Response
This manuscript investigated the relationship between allergic component of serum and appendicular lavage fluid IL-5 levels, eosinophilic infiltration at appendicular wall and the aetiology of appendicitis and concluded that the hypothesis of a local allergic component, hypersensitivity type I reaction, in acute phlegmonous appendicitis. It is very interesting, but there were some major comments.
- If the occurrence of acute appendicitis is related to type I allergy, why only focus on the study of IL-5 level?
In a previous work, we have evaluated the Th2 cytokines and IgE involvement in acute appendicitis (References 5 and 6). The aim of this work was to evaluate the local and the systemic IL-5 and eosinophilic responses in acute appendicitis and so, our focus was only in IL-5 and eosinophils, both at local level (Appendicular lavage fluid) and at systemic level, in blood.
- If acute appendicitis is caused by allergy, why does the patient have no other allergic symptoms?
The digestive tract is in contact with thousands of antigens during life time. The antigen milieu in appendix associated with all the kind of cells involved in hypersensitivity reaction type I are present in the appendix, and so, allergy is a specific local phenomenon. Allergy is a complex event. Frequently people with allergy have more than one allergic disease, but it is not unusual the presence of just one allergic disease. Furthermore, the first manifestation of allergy can be, for example, an anaphylactic reaction, and so, acute appendicitis can be the first manifestation with clinical relevance to an unknow antigen.
- Is eosinophil infiltration in the appendix wall the result or cause of appendicitis?
This is a very interesting question. In allergy, there is local release of IL-5 that acts as a chemotactic factor for eosinophils, that invade the target organ, in the present case, the appendices Being so, cytokines may act as a trigger for appendicitis. Eosinophils release several cationic proteins, such as Eosinophil cationic protein, eosinophil peroxidase or eosinophil derived neurotoxin, that have a powerful inflammatory and cellular destructive properties. it seems that eosinophil at appendicular wall, act as a promotor of the pathologic consequences characteristic of the different phases of appendicitis.
- Number of eosinophils in appendicitis was increased. Is there any increase of eosinophils in the terminal ileum or cecum?
This is an important and pertinent issue. If we could have samples from the terminal ileum and cecum the answer was straight forward and would contribute for the understanding of the involving milieu in a wide segment of bowel, surrounding the appendices. To perform colonoscopy with biopsies is not contemplated in good practice guidelines and would rise ethical issues.
- The lavage solution of isolated appendix cavity by surgery could not completely reflect the situation in vivo.
Local inflammatory changes at the lung are clinically evaluated with bronchoalveolar fluid samples, a procedure that has been performed for many years. We develop the concept of appendicular lavage fluid, hopping that the harvested fluid will reflect, as possible, the inflammatory environment of the appendix. In the context appendicitis, the lavage solution, in fact, similarly to alveolar lavage, should reflect the main changes occurring in the appendix.
- According to the experimental design of this article, it is difficult to conclude that the occurrence of type I allergic reaction was the etiology of acute appendicitis.
Thank you for all question. Our results point to a meaningful role of Type I allergic reaction during the evolution of the sequential phases of appendicitis, and they are probably involved in the early stage of appendicitis in a substantial number of cases.
Most probably the aetiology of acute appendicitis is multifactorial triggered by different conjugations of the appendix ecosystem.
Reviewer 3 Report
well stactured paper but with the use of figures from privious papers (your own articles) eg. "Increased IgE Deposition in Appendicular Tissue Specimens Is Compatible with a Type I Hypersensitivity Reaction in Acute Appendicitis"
it is beter to avoid these.
you must correct, change or discharge this error.
Furthermore, all of your papers (the present and references 5, 6, 7 [self-citations]) could be one excellent paper (it is better to avoid this kind of "cutting")
Author Response
well stactured paper but with the use of figures from privious papers (your own articles) eg. "Increased IgE Deposition in Appendicular Tissue Specimens Is Compatible with a Type I Hypersensitivity Reaction in Acute Appendicitis"
it is beter to avoid these.
you must correct, change or discharge this error.
Furthermore, all of your papers (the present and references 5, 6, 7 [self-citations]) could be one excellent paper (it is better to avoid this kind of "cutting")
Thank you for your commentaries
The figure is from a previous study, where we used the same methodology for counting IgE stained cells (Reference 6). The content of the figure has now been inserted in the test (6) and the figure was discharged.
The present article is part of a project where we are studying the possibility of an allergic contribution in the aetiology of acute appendicitis. In fact, the present article could be integrated in the previous one, but at that time, our focus was on Th2 cytokines contribution and eosinophils presence in appendicitis was a more recent aspect of the investigation.
Round 2
Reviewer 2 Report
Thanks to the editor for this opportunity to review the revised manuscript number ijms-1943893. Thanks to the authors for making required reply point by point and for their efforts to revise and address the raised concerns. However, it is suggested to modify the prescription of conclusion , and to add the relevant content in the discussion according the author's reply and the reviewer's comment.
Author Response
Dear Reviewer 2
I am very grateful for the pertinent questions you raised and which were integrated into the text, which greatly benefited and improved the quality of the article.
I hope the answer to your questions is as intended.
Best Regards